# Strengthening Community End-of-Life Care through Implementing Measurement-Based Palliative Care

**DOI:** 10.3390/ijerph19137747

**Published:** 2022-06-24

**Authors:** Margaret H. Sandham, Emma Hedgecock, Mevhibe Hocaoglu, Celia Palmer, Rebecca J. Jarden, Ajit Narayanan, Richard J. Siegert

**Affiliations:** 1School of Clinical Sciences, Auckland University of Technology (AUT), North Shore Campus, 90 Akoranga Drive, Northcote, Auckland 0627, New Zealand; richard.siegert@aut.ac.nz; 2Oceania Healthcare, Waikato 3420, New Zealand; emma.hedgecock@oceaniahealthcare.co.nz; 3Cicely Saunders Institute of Palliative Care, Florence Nightingale Faculty of Nursing, Midwifery and Palliative Care, King’s College, London WC2R 2LS, UK; mevhibe.hocaoglu@kcl.ac.uk; 4Hospice West Auckland, Te Atatu, Auckland 0610, New Zealand; celiap@hwa.org.nz; 5Department of Nursing, Melbourne School of Health Sciences, 161 Barry Street, Carlton, VIC 3053, Australia; rebecca.jarden@unimelb.edu.au; 6School of Engineering, Computing and Mathematical Sciences, Auckland University of Technology (AUT), AUT Tower, 2-14 Wakefield Street, Auckland 1010, New Zealand; ajit.narayanan@aut.ac.nz

**Keywords:** measurement-based care, palliative care, patient-reported outcome measures

## Abstract

The increasing demand for palliative care in New Zealand presents a potential threat to the quality of service delivery. One strategy to overcome this is through the implementation of valid and reliable patient-reported outcome measures. This mixed-methods study aimed to (1) implement measurement-based palliative care (MBPC) in a community palliative care service in Auckland, New Zealand; (2) evaluate the clinical utility of MBPC perceived by clinicians; (3) describe patient characteristics as measured by the Integrated Palliative Care Outcome Scale (IPOS), the Australasian Modified Karnofsky Performance Scale (AKPS), and Phase of Illness (POI); and (4) evaluate the internal consistency of the IPOS. Participants were over 18 years of age from a community outpatient palliative care service. In a phased approach to implementation, healthcare staff were educated on each instrument used for patient assessment. Uptake and internal consistency were evaluated through descriptive statistics. An interpretive descriptive methodology was used to explore the clinical utility of MBPC through semi-structured interviews with seven clinical staff members. Individual patient assessments (*n* = 1507) were undertaken predominantly on admission, with decreasing frequency as patients advanced through to the terminal phase of their care. Mean total IPOS scores were 17.97 (SD = 10.39, α = 0.78). The POI showed that 65% of patients were in the stable phase, 20% were in the unstable phase, 9% were in the deteriorating phase, and 2% were in the terminal phase. Clinicians reported that MBPC facilitated holistic and comprehensive assessments, as well as the development of a common interdisciplinary language. Clinicians expressed discomfort using the psychosocial and spiritual items. Measurement-based palliative care was only partially implemented but it was valued by staff and perceived to increase the quality of service delivery. Future research should determine the optimal timing of assessments, cultural responsivity for Māori and Pacific patients, and the role of MBPC in decision support for clinicians.

## 1. Introduction

The demand for palliative care services in New Zealand is forecast to increase by 51% over the next 20 years with increasing comorbidities, longer life expectancies, and new treatment and palliation options, making palliative care more complex [1]. Within developed countries with aging populations, close to three quarters of older adults live with disability and chronic diseases [2]. A simulation model of UK primary care patients projected a near doubling of patients with four or more diseases between 2015 and 2035, and predicted that two-thirds will have cognitive impairment (without dementia), depression, or dementia [3]. This presents challenges to the quality of palliative care services, especially when they are increasingly being delivered by non-specialist primary care, aged residential care, and hospital providers [1,4]. Communication between services and within teams can be hindered through the lack of specialist assessment skills, a lack of information and standardized language to quantify symptoms, and their intensity. The use of patient-reported outcome measures (PROMs) is a key strategy in overcoming threats to the delivery of quality palliative care services. Strong evidence has been found in studies that have evaluated the impact of PROM collection in clinical settings for enhanced patient–clinician communication, the identification of unrecognized symptoms, the increased monitoring of symptoms, improvements in patient experiences and satisfaction, and increased clinician action from symptom reports [5]. Although a key action for increasing the quality of end-of-life care service delivery, a core set of outcome measures for palliative care in New Zealand has not yet been identified and evaluated [1,6].

Measurement-based care (MBC) is the repeated and systematic administration of empirically supported PROMs to guide clinical decision making in individual patient care and to demonstrate the value of treatments [7]. The repetition of outcome measures throughout treatment produces improved outcomes for patients when compared to a single screening measure with no systems in place to monitor treatment response [8]. Multiple benefits of MBC have been reported in mental health service delivery such as: (1) the early identification of issues to guide clinical decision making [9,10]; (2) the demonstration of the value of services to funding bodies [7]; (3) flexibility for clinicians to deliver care from their diverse clinical experience and across settings [10]; (4) facilitation of the development of treatment algorithms; (5) improvement in the detection of residual symptoms; and (6) team-based collaborative care using standardized language [7]. The use of MBC in mental health service delivery has resulted in superior outcomes compared to treatment as usual and these outcomes transcend patient diagnosis and clinician factors [9]. The evidence for MBC is strong enough that Scott and Lewis [10] argue MBC could be a minimal intervention used to produce a substantial impact on patient outcomes when compared to more complex and burdensome evidence-based practices.

Measurement-based palliative care (MBPC) translates the best available evidence from mental health service delivery to address priorities in New Zealand palliative care [1,4,6]. An evaluation of the national implementation of the Palliative Care Outcomes Collaboration (PCOC) in Australian palliative care service delivery showed improved symptom control, increased understanding of the prevalence of conditions, enhanced service planning, and international benchmarking [11,12]. The use of standardized PROMs enables clinicians, independent of their background or skill level, to provide a high-quality assessment [13]. Measurement-based palliative care may overcome challenges to nursing-sensitive patient outcomes that previous authors [13,14] identified through variations in the quality of the nursing assessment, nursing diagnosis, and subsequent response. 

Identifying, evaluating, and implementing a core set of outcome measures is a priority for New Zealand palliative care [1,6]. The present mixed-methods study aims to (1) implement MBPC in a community palliative care service in Auckland, New Zealand; (2) evaluate the clinical utility of MBPC perceived by clinicians; (3) describe patient characteristics as measured by the Integrated Palliative Care Outcome Scale (IPOS), the Australasian Modified Karnofsky Performance Scale (AKPS), and Phase of Illness (POI); and (4) evaluate the internal consistency of the IPOS.

## 2. Materials and Methods

### 2.1. Study Setting

Data collection took place at a community palliative care service which provides services to approximately 230 patients living in a mixed urban and rural area in Auckland, New Zealand. Approximately 25% of patients were under 65 years of age. The ethnic composition of the Auckland region is 53.5% New Zealand European, 28.2% Asian, 15.5% Pacific Island, and 11.5% Māori, and the service covers areas of socioeconomic deprivation [15]. The present study occurs in the context of a broader governmental directive to identify and implement a core set of outcome measures in New Zealand palliative care services. Individual private care providers and public services currently undertake implementation of outcome measures ad hoc, determining the measures that best meet their service needs. Consequently, the present study was undertaken in part to provide information to other Hospice New Zealand services to support the implementation of MBPC.

### 2.2. Design

The present mixed-methods study is part of a program of research investigating MBPC [16,17,18]. Here we report (1) the practical implementation of MBPC within the clinical setting, (2) an interpretive descriptive qualitative study focused on the clinical utility perceived by clinicians, and (3) a longitudinal observational study describing patient characteristics as measured by both patient-reported outcome measures and clinician observer-reported measures.

### 2.3. Participants

Patients were included if they were over 18 years of age and were able to complete the study measures independently, with help or by proxy. Patients were excluded if they had very limited English language and/or moderate to severe cognitive impairment, and/or if the clinical team judged them as being too unwell or distressed to participate. Staff were eligible for interviews if they had participated in the implementation of MBPC and had used the measures with patients in the previous six-weeks.

### 2.4. Ethical Considerations

Patients as part of their admission signed a release to use their de-identified data for quality and auditing purposes; therefore, separate consent for the study was not sought. However, external ethical approval was sought from the New Zealand Health and Disability Ethics Committee, and the study was assessed to be low-risk and an exemption was granted. Ethical approval for staff interviews and an evaluation of patient data were obtained from the board of ethics as part of Hospice New Zealand.

### 2.5. Study Measures

Three measures were used. (1) The IPOS [19] is a ten-item measure of symptom burden in the domains of psychosocial, spiritual, social, and physical symptoms. Symptoms were assessed on a 0 to 4 Likert scale (ranging from ‘not at all’ to ‘overwhelmingly’), and two open-ended questions had a free-response option so that patients could record other symptoms of importance to them. An IPOS score of 0 or 1 requires less clinical intervention compared to a score of 3 or 4 which requires high levels of intervention. A score of 0 indicates that the patient has no impact from the symptom, 1 indicates a slight impact but minimal distress, whereas a 3 or 4 indicate an overwhelming impact of the symptom [20]. The IPOS is a reliable and valid measure that has been used extensively internationally and has been translated into 14 languages [19,21,22,23]. The IPOS shows promise for implementation within New Zealand palliative care but at the time of this study had not yet been evaluated in New Zealand. (2) The Australasian Karnofsky Performance Scale—AKPS [24] is a clinician-administered rating scale based on observations of the patient’s performance on three dimensions of work, activity, and self-care. Patients were allocated a percentage score from 100% (normal, no complaints, no evidence of disease) to 0% (dead). Clear criteria were provided for increments of 10%. The AKPS has demonstrated good psychometric properties in Australian palliative care populations [24] and is currently in use within New Zealand. (3) The Phase of Illness—POI [25] classifies patients into one of five distinct stages of illness (stable, unstable, deteriorating, terminal/dying, deceased) according to their care needs. Phases are non-sequential (patients can move back and forward through phases one to four), and a change in phase signals a requirement to re-assess the patient and family’s needs. POI has been validated in an Australian sample of 1317 palliative care patients and showed validity in capturing information related to clinical need [26]. The POI has acceptable interrater reliability [18,25]; is unidimensional; does not show differential item functioning by age, sex, and ethnicity [27]; and is in use in New Zealand.

### 2.6. Procedure

#### 2.6.1. Phase One: Implementation

A staged implementation took place between December 2017 and September 2019 to increase staff engagement in the change of practice. To reduce the perceived time burden on staff, the AKPS and POI were initially introduced in place of other documentation at the patient admission assessment, followed by the IPOS. The IPOS was then increased to be undertaken at six weekly intervals or when clinically indicated, such as at a change of phase identified by the POI, with a minimum of a one-week interval for patients who were rapidly moving between phases one to four. The focus was on embedding MBPC into clinical practice rather than it being seen as merely an auditing tool.

Previous research has identified the importance of education and the presence of a coordinator during implementation [28]. No specific role was created for implementing MBPC; however, education and coordination around MBPC were provided by the clinical nurse specialist (author E. H) and the senior medical officer (author C.P). Clinicians involved in the implementation were aware of the concurrent research project and that they could choose to participate in a later qualitative stage. Education included the rationale for using outcome measures, how and when to administer the measures, and guidelines on the interpretation and documentation of the measures. A weekly meeting was held where patient outcome measures were discussed and interpreted as a team, with discussions on scoring held to enhance interrater reliability.

#### 2.6.2. Phase Two: Evaluation

The palliative care service led the evaluation of service provider utility through developing an interview guide (see Appendix A) and employing an independent research assistant to undertake confidential structured interviews with staff who had begun to use the IPOS, the AKPS, and POI in their patient care. The participants were seven nursing and social work staff recruited internally within the organization through advertisements. The research assistant was a trainee psychologist who had no prior experience of palliative care but was trained in qualitative research methods. The semi-structured interviews had a guide to elicit perceptions of service provider utility, patient and clinician benefits, and applicability to New Zealand models of health. The interviews were audio-recorded and transcribed. An interpretive descriptive approach [29] was chosen, allowing the participants’ experiences to be integrated with the authors’ clinical knowledge. The interviews were transcribed verbatim, and common themes and patterns were identified by two researchers, ensuring that the data were examined in their original context. The interpreted data were presented back to the staff within the service provider to determine the alignment with their experiences.

The uptake of MBPC was evaluated through identifying the proportion of patients assessed using the IPOS, the frequency of reassessments, and longitudinal changes in the usage of the IPOS.

### 2.7. Analysis

Descriptive statistics of means and percentages were used. Cronbach’s α was used to assess the internal consistency reliability of the IPOS items. Cronbach’s α statistic is a correlation of items which shows how closely related the items are on the measure. Values greater than 0.7 are considered good, 0.8 are considered substantial, and 0.9 are considered excellent [30]. 

## 3. Results

### 3.1. Uptake

Data were collected between December 2017 and September 2019. A total of 1507 separate assessments were made based on 804 patients. Ages ranged from 23 to 101 years (M = 70.9, SD = 13.7; Mdn = 73.0, IQR = 20), as shown in Table 1.

Questionnaires were completed upon admission and then subsequently between one to three months, with 498 patients (61.9%) on admission only, 131 (16.3%) on admission and one further occasion, 74 (9.2%) on two further occasions, 41 (5.1%) on three further occasions, and the remaining 60 (7.5%) on four or more further occasions. Questionnaires were mainly completed upon admission and then subsequently between one to three months, averaging 1.87 assessments per patient. Most questionnaires were administered to stable patients, and fewer patients in the later phases received assessments.

### 3.2. Sample Characteristics

The sample was predominantly representative of the local Auckland population with European (54%), Pacific (12%), and Māori (12%) ethnicities proportionally represented. The exception was those of Asian ethnicity (6%), who were underrepresented against the general Auckland population [15].

Patients were primarily diagnosed with oncological cancer 569 (70.8%), 140 (17.4%) with organ failure (heart, lung, liver, kidneys), 38 (4.7%) with haematological cancer, 24 (3%) dementia (including Alzheimer’s and Parkinson’s), 16 (2%) with other neurological (motor neurone disease, multisystem atrophy, supranuclear palsy), and 17 (2.1%) with acute event (stroke, sepsis, abdominal aortic aneurysm). 

### 3.3. Sample Assessment Using Scales

Of the 1507 assessments, AKPS scores were available for 1439 (M = 59.4, SD = 17.8; Mdn 60, IQR = 20; 68 missing). Table 2 reports the breakdown of IPOS and AKPS scores, both raw and aggregated.

When AKPS scores for patients with more than one assessment were averaged, the mean reduced to 56.5 (SD = 18.6; Mdn = 60, *n* = 767, 37 missing), indicating that patients on average could care for their personal needs with occasional assistance. The AKPS lower quartile score was 50% (requires considerable assistance and frequent medical care), the middle quartile score was 60% (requires occasional assistance, can care for most personal needs), and the upper quartile score was 70% (cares for self; cannot carry out normal activities or do active work). Table 3 reports descriptive statistics for all three assessments undertaken (AKPS, POI, and IPOS). 

All 1507 assessments included a total IPOS scale score, with the mean score calculated to be 10.52 (SD = 6.06; Mdn = 10, IQR = 8.0), indicating that patients reported minimal impact of symptoms and required lower levels of clinical intervention. The lower quartile score was 6, the middle quartile was 10, and the upper quartile was 14. After averaging for patients with more than one assessment, an increase in symptom impact was evident, and the mean increased to 11.18 (SD = 5.59; Mdn = 10.5), with a lower quartile score of 7.0, a middle quartile score 10.76, and an upper quartile score 14.24 (*n* = 804). Phase of Illness was available on 1453 of the 1507 separate assessments (54 missing), with 980 (65.0%) in the stable phase, 301 (20.0%) in the unstable phase, 138 (9.2%) in the deteriorating phase, and 34 (2.3%) in the terminal phase.

The mean AKPS across phases was 56% (19% = SD), with the highest score in the stable phase (63%), though it reduced at each phase, showing deterioration. The highest IPOS scores were observed across weakness, poor appetite, and poor mobility items in the unstable, deteriorating, and terminal phases. Although higher than some other items, pain scores stayed relatively stable. For a more detailed account of the statistical analysis interested, readers are invited to refer to [17] for more information.

### 3.4. Internal Consistency

Cronbach’s α of the IPOS items across all 1507 questionnaires was 0.78, with highest internal consistency when patients completed with the help of family/friends (0.83), followed by on their own (0.80) and with the help of staff (0.74).

### 3.5. Staff Feedback

#### 3.5.1. Bringing Patients Holistic Needs to Light

Participants reported that the structured way in which they spoke with patients when using the IPOS allowed them to be holistic and comprehensive in their assessment and decrease the dominance of physical symptoms in their discussions.

*Initially when I started using it I…I was a wee bit sceptical because I thought its very prescriptive and as a clinician I don’t tend to like that. I like to be more autonomous. But then when I started using it I could see that there are certain things in here that I might have missed had I not had this*.Nurse 2.

*I do like the sort of assessment that’s coming from patients and family members, so I quite like it…this is solidly patient centred I feel. I think it gives them the option to express themselves holistically and also normalises these questions…. It will definitely benefit them in terms of being asked of the questions that’s not just on body parts, or psychosocial part, but both parts*.Nurse 5.

Participants reported increased awareness in their patients’ individualised needs and were confident that their care delivery had become more holistic and enhanced.

#### 3.5.2. Participant-Reported Utility of PROMS

Participants felt that the PROMs were easy to use and provided structure to their visit and consistency among their patient assessments. The adaptability of the IPOS with the use of free-response sections was helpful for staff. However, the repetition of the IPOS at changes of phase took longer, perhaps due to the emphasis on embedding the assessments at intake.

*So a good example is we have done IPOS on a client 3 weeks ago. We have met the client again, and the IPOS has had changes on it. And based on those changes we have been able to implement a care plan directed on those changes*.Nurse 3.

#### 3.5.3. Making Emotional Wellbeing Assessments Routine

Participants reported that they would often allow physical symptoms to dominate the assessment, but the structure of the IPOS made asking questions on mental health routine. Participants observed that patients’ mental health became a shared responsibility within the interdisciplinary team.


*Personally I’d always be a little bit more kind of concerned about asking like depression and all of them questions but, having them on a piece of paper and reading them out sometimes is helpful that it’s … direct and its part of an assessment, so it kind of keeps it straight forward ….*
Nurse 6.


*So I kind of feel patients would feel—“okay I’m not just being asked because I’m sick and that’s why I’m appearing as depressed person, but …oh okay, it’s a normal question that everyone will be asked” okay we’ll just talk about it… and maybe just opens up the space...*
Nurse 5.

While the ability to raise psychosocial aspects such as depression was valued by participants, some felt that these questions may be perceived by patients as an intrusion and that they were ill-equipped to respond should a patient reveal psychological distress.

*I’m not sure about some of the other questions on the other side of the page whether that’s, I sometimes feel like I’m…but that could be me… going and asking personal stuff. For a complete stranger going into somebody’s house saying have you got peace, you know it’s… I’m not sure if its right*.Nurse 1.

#### 3.5.4. Uncertain Cultural Applicability

Participants emphasised that the cultural responsivity of the assessment tools was essential, but some lacked awareness of Māori health models to appraise the IPOS.


*We particularly look at Te Whare Tapa Wha as important that were looking at the physical but also the social and emotional, spiritual, family—and the fact that it does mention family, whanau is really important because at the end of the day, that’s usually what people put as number one as their concern that they’re being a burden to their family or that their family is going to be the ones left behind when they go.*
Nurse 4.

Participants thought that the IPOS was applicable for the New Zealand cultural context, though they did not expand on their rationale or link it to indigenous Māori or Pacific models of health.

#### 3.5.5. Developing a Shared Clinical Language

Participants reported that the implementation of MBPC changed the clinical language used and improved communication between clinicians.


*It’s made it easier to assess people, our patients, like it gives us something to go by and it kind of ensures we don’t miss anything and we don’t miss any questions that are important at that time so it’s definitely improved that. From what I see in handovers and stuff its quite helpful for the medical officers to keep an eye and, they kind of base their assessment on these as well, so it kind of continues. We can see straight out what one score is and what it is on the next one so we can see quite quickly if there is a difference or if there’s any improvement or if things might have deteriorated even further.*
Nurse 6.

Furthermore, IPOS and AKPS categories were commonly used when communicating patient status within the organisation and when requesting general practitioner interventions. Clinicians reported that using the IPOS helped them to provide a higher standard of care than previously.

## 4. Discussion

The present study described the implementation of MBPC in a community palliative care service, including the characteristics and illness severity of the patients in the service using the IPOS, the AKPS, and POI. Finally, we reported the perspectives of the multidisciplinary users of the utility of these outcome measures. The majority of MBPC assessments were completed on intake, with a tapering off on repeated measures. Clinicians reported the value of embedding the IPOS into their routine care of patients, resulting in enhanced communication both within the interdisciplinary team and with the patients themselves, thereby revealing physical and psychosocial symptoms. Nursing and medical leadership working together enthusiastically facilitated the interdisciplinary team to find creative ways of embedding MBPC into routine practice.

The median score of 60 for the AKPS is in line with previously reported scores in a palliative care setting [23]. The average AKPS score of 57 across all patient records indicates that patients in this cohort only required occasional assistance. Similar scores were observed in pain across the phases, suggesting that good pain control may have been achieved. However, the weakness/lack of energy, poor appetite, and poor mobility items showed change over phases, suggesting that these symptoms may be more difficult to address. 

The IPOS demonstrated good internal consistency and reliability, with average Cronbach’s α = 0.78 slightly higher than previous staff-rated α coefficients [31]. Internal consistency was improved when patients completed the questionnaire with the help of family or friends (α = 0.83) compared to independently (α = 0.80) or with the help of staff (α = 0.74). As the median AKPS score for this cohort is in line with studies elsewhere, the increased reliability when family supported the assessment could be due to the New Zealand context, where whānau (extended families or community of related families) are encouraged to actively participate in caring for dying relatives [32]. Increased reliability in scoring may relate to patients being encouraged by whānau to express their experiences and feelings in more detail than through self-reflection or staff assistance.

The theme of “making emotional wellbeing assessments routine” suggests that implementing MBPC may in part enable clinicians to break the ‘conspiracy of silence’ where neither patient nor clinician initiates discussion of sensitive matters [33]. Within New Zealand, the conspiracy of silence may also extend to difficulty initiating conversations in the essential but often unspoken area of wairuatanga (spirituality) [32,34]. Participants reported that IPOS broadened their assessments from focusing on physical symptoms to psychosocial and mental health concerns. However, participants experienced discomfort in discussing psychosocial and spiritual matters, and requested further education. The necessity of ongoing staff education and support to fully utilize the potential of PROMs in the present project aligns with experiences reported elsewhere [28]. We recommend that staff should undergo additional education to address knowledge gaps in mental health, as well as spiritual and cultural assessment.

The high number of completed patient admission assessments supported the perceived value of PROMs. However, the frequency of repeated measures decreased when patients required higher levels of care (e.g., unstable, deteriorating, and terminal POI). Clinical staff resourcing in New Zealand has been found to reduce the use of PROMs despite their perceived value [35]. The optimal timing of assessments should be investigated, as phase change offers an identifiable marker for clinicians. On the other hand, when patients are frequently changing condition, repeated assessments may become burdensome.

Although the present study sought information from clinicians on the appropriateness of the measures for Māori, we did not find enough evidence to support or discourage the use of these measures for Māori and Pacific people. Future research should aim to engage cultural partners as co-investigators. Cognitive interviewing, e.g., Murtagh, Addington-Hall, and Higginson [36], may identify content validity and cultural acceptability issues for Māori and Pacific people.

The qualitative component of this mixed-methods study was limited by a small sample of self-selected available participants, even though findings were presented back to staff who supported the validity. The patient data set was heterogeneous regarding disease, with varying numbers in each category and a preponderance (over 70%) in oncological cancer. Furthermore, patients in the deteriorating and dying phase constitute a relatively small 11.5% of the cohort, leading to possible bias in results towards patients in the other two phases. The COVID-19 pandemic has occurred since data collection took place, consequently affecting the staff experiences reported and, in turn, the generalizability of the findings to the post-pandemic healthcare setting. 

Understanding the benefits and key educational gaps when implementing measurement-based palliative care enabled nursing leaders to identify areas to deliver education and improve the success of implementation. For example, we found initial evidence of discomfort and a lack of knowledge of clinical terms of mental health assessment. This highlights the benefit of a structured clinical assessment to prompt areas for assessment that may otherwise be avoided and increase consistency between staff. Implementing MBPC facilitates a common language used during interdisciplinary meetings or between-service collaborations in patient care, which was valued by staff and thought to increase the quality of service delivery. In summary, implementing MBPC strengthened community end-of-life care by: (a) instituting a set of internationally recognized measures and demonstrating their reliability; (b) reporting a benchmark or reference score for a large NZ community patient sample; and (c) providing qualitative evidence that clinical staff found these measures acceptable and informative and could improve aspects of care that are sometimes neglected.

## 5. Conclusions

The implementation of MBPC was a meaningful practice change in a community-based palliative care setting. Clinicians reported increased awareness of patients’ holistic needs and perceived an enhancement of quality care. Asking psychological and spiritual questions raised discomfort for some clinicians, but brought attention to the need for mental health assessment. This clinician discomfort highlights the need for additional clinician training to increase confidence in responding to psychosocial needs identified using the IPOS. Using MBPC assists both clinicians and patients to discuss sensitive issues and break the conspiracy of silence where issues are not raised by either the patient or the clinician. 

The IPOS demonstrated good reliability and internal consistency in the present context and this was enhanced when completed with the support of family or friends. Future research should explore (a) optimizing the timing of administration, (b) how changes in AKPS and IPOS interact with POI, and (c) how all three measures may be used together in decision support across services.

## Figures and Tables

**Table 1 ijerph-19-07747-t001:** Sample characteristics.

Characteristics	*N*	%
Female	404	50.2
Male	398	49.5
Unreported gender	2	0.3
*Ethnicity*		
Caucasian	435	54.1
Māori	100	12.4
Pacific	95	11.8
Asian	52	6.5
Other	122	15.2
*Diagnosis*		
Oncological cancer with organ failure (heart, lung, liver, kidney)	569	70.8
Haematological cancer	140	17.4
Dementia	24	3.0
Other neurological disease (motor neurone, multisystem atrophy, supranuclear palsy)	16	2.0
Acute event (stroke, sepsis, abdominal aortic aneurysm)	17	2.1
*Completion*		
Independent	420	27.9
Help of staff	596	39.5
Help of family/friends	132	8.8
Not known	359	23.8
*Time of assessment*		
Admission	498	61.9
Admission + one other occasion	131	16.1
Admission + two other occasions	74	9.3
Admission + three other occasions	41	5.2
Admission + four other occasions	60	7.5
*Total assessments* (*N* = 804 patients)	1507	

**Table 2 ijerph-19-07747-t002:** Quartile breakdown of all IPOS and AKPS scores and after aggregation (A) by patient.

	IPOS	IPOS A	AKPS	AKPS A
Valid	1256	672	1439	767
Missing	251	132	68	37
25	6.00	7.00	50.00	50.00
50	10.00	10.50	60.00	60.00
75	14.00	14.00	70.00	70.00

Note: The Integrated Palliative Care Outcome Scale (IPOS), the Australasian Modified Karnofsky Performance Scale (AKPS), aggregation by patient (A).

**Table 3 ijerph-19-07747-t003:** Descriptive statistics for the AKPS, IPOS, and POI.

Item	All		Phase of Illness (POI)			
	Mean	Std. Dev.	Stable	Unstable	Deteriorating	Dying
*IPOS*						
Pain	1.40	1.13	1.25	1.82	1.66	1.68
Shortness of breath	1.19	1.14	1.09	1.34	1.46	1.11
Weakness or lack of energy	1.96	1.09	1.77	2.22	2.53	2.89
Nausea	0.53	0.88	0.43	0.84	0.61	0.53
Vomiting	0.23	0.62	0.17	0.39	0.34	0.29
Poor appetite	1.20	1.20	0.97	1.59	1.83	2.47
Constipation	0.84	1.10	0.73	1.00	1.12	1.00
Sore or dry mouth	0.90	1.02	0.81	1.02	1.13	1.44
Drowsiness	1.00	1.05	0.88	1.10	1.42	1.95
Poor mobility	1.45	1.22	1.27	1.67	2.07	2.95
Total IPOS score	10.52	6.06	9.24	12.99	13.75	15.62
*AKPS*						
AKPS%	59.44	17.83	63.32	58.89	44.71	12.06
Score (A)	11.18	5.56				
AKPS% (A)	56.50	18.65				

Notes. Breakdown of Integrated Palliative Care Outcome Scale (IPOS) items (range 0–4), overall IPOS score and Australasian Karnofsky Performance Scale (AKPS) % score in totality and by Phase of Illness (POI) for all patient records (*n* = 1507; stable *n* = 980; unstable *n* = 301; deteriorating *n* = 136; terminal *n* = 34), with aggregated IPOS (A) and AKPS % (A) (*n* = 672 and *n* = 767 patients, respectively) in the bottom two rows. Cronbach’s α total IPOS (α = 0.78), with the help of family/friends (α = 0.83), patient self-report (α = 0.80), with the help of staff (α = 0.74).

## Data Availability

Data informing the findings of this study are available from the corresponding author on reasonable request.

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
