# Peer review of "Strengthening Community End-of-Life Care through Implementing Measurement-Based Palliative Care"

_ijerph, 2022, doi:10.3390/ijerph19137747_

Round 1
Reviewer 1 Report
Dear authors:
I have reviewed the manuscript entitled “Strengthening Community End-of-Life Care through Implementing Measurement-based Palliative Care”. It is an interesting study that is very useful in supporting decision-making by health professionals in the context of palliative care. It has a pleasant and easy-to-read presentation. The appropriate methodology for the type of study. I believe readers of the journal with a special interest in palliative care will appreciate your work as I have.
On a few notes to further develop the manuscript, I would suggest:
- The implementation phase occurs during 2017 and 2019 (pré-covid time), There is already a 3-year time gap, between implementation and this manuscript, that interval of three years, during covid, which many social and sanitary changes were experienced, interfered with the mental health of patients and health professionals caring for people with palliative needs. So, it is suggested, that fact could be mentioned in conclusion as a study limitation, and a recommendation for a new implementation in period after covid appearance, because the mental health should have new concerns and could interfere in scales outcomes;
- For the interview guide carried out is referred to be available in supplementary information, but they are not available with the manuscript;
- line 3 from page 3 there is reveled the setting “Hospice West Auckland” I believe it is a direct mention of the study setting, so if that is right, I suggest to make change it, to an indirect mention, in order to guarantee ethical assumptions;
-The cited references in the last five years are ≈30% it could be interesting have most recent references;
I have nothing to add and I wish you all the best with its publication.
Author Response
|
Reviewer 1 Comment 1: I have reviewed the manuscript entitled “Strengthening Community End-of-Life Care through Implementing Measurement-based Palliative Care”. It is an interesting study that is very useful in supporting decision-making by health professionals in the context of palliative care. It has a pleasant and easy-to-read presentation. The appropriate methodology for the type of study. I believe readers of the journal with a special interest in palliative care will appreciate your work as I have.
|
Authors’ response 1: Thank you for your support in highlighting some of the strengths of our study and recommendations for enhancing our manuscript. |
|
Reviewer 1 Comment 2: The implementation phase occurs during 2017 and 2019 (pré-covid time), There is already a 3-year time gap, between implementation and this manuscript, that interval of three years, during covid, which many social and sanitary changes were experienced, interfered with the mental health of patients and health professionals caring for people with palliative needs. So, it is suggested, that fact could be mentioned in conclusion as a study limitation, and a recommendation for a new implementation in period after covid appearance, because the mental health should have new concerns and could interfere in scales outcomes; |
Authors’ response 2: We have implemented this suggestion and added a line in limitations “The COVID-19 pandemic has occurred since data collection took place, consequently this may affect the staff experiences reported and in turn the generalizability of the findings to the post-pandemic healthcare setting.” Page 12 line 477 |
|
Reviewer 1 Comment 3: For the interview guide carried out is referred to be available in supplementary information, but they are not available with the manuscript |
Authors’ response 3: We are grateful for the reviewer to note this omission, we have included it as an attachment to the response to the email. |
|
Reviewer 1 Comment 4: line 3 from page 3 there is reveled the setting “Hospice West Auckland” I believe it is a direct mention of the study setting, so if that is right, I suggest to make change it, to an indirect mention, in order to guarantee ethical assumptions;
|
Authors’ response 4: Removed reference to the name of the study setting. Although Hospice West Auckland had given consent for the study location to be acknowledged, we have removed this. |
|
Reviewer 1 Comment 5: The cited references in the last five years are ≈30% it could be interesting have most recent references |
Authors’ response 5: we have included newer references and removed some of the older ones. A number of the references relate to the primary publications developing the outcome measures, consequently they will be older. |
Reviewer 2 Report
The manuscript illustrates the implementation of MBPC in a community palliative care service, including the characteristics and illness severity of the patients in the service using the IPOS, AKPS, and POI. Also, the Authors report the qualitative evaluations of participants to the survey and of clinicians who integrated IPOS into their routine patients care .
The data were collected between December 2017 and September 2019. A total of 1507 assessments were carried out on 804 patients. Descriptive statistics of means and percentages are used to interpret the results. Cronbach’s α is used to assess the internal consistency of the IPOS items. Questionnaires were mainly completed at the time of admission and subsequently between one to three months, with an average of 1.87 assessments per patient. Of the 1507 assessments, AKPS scores were available for 1439. Cronbach’s α of the IPOS items across all 1507 questionnaires was 0.78.
Comments
The manuscript is well written and illustrates an interesting research that could introduce some novelty in the field of palliative cares. However, the literature examined is rather dated (only one third of the references has been published less than five years ago).
Quantitative analyses could also be improved, as they are based only on descriptive statistics and results presented are not clear. As an example, it is not clear why absolute frequencies reported in table 1 concerning diagnoses, completion, and time of assessment do not sum to 1.507.
Furthermore, data illustrated in Table 2 are not commented on the main body of the manuscript, or, at least, it is not clear to me how they relate to the data illustrated in Table 3 and commented on lines 241-250. I could not find data concerning POI (lines 248-250) in Tables 1-2.
Summary statistics illustrated in Table 3 are difficult to interpret, consequently Authors are invited to improve the interpretation of the data in the main body of the manuscript. As an example, why both St.Dev. e St.Error are reported?
Finally, the analysis of Cronbach’s α of the IPOS items (lines 271-273 could be integrated with a table illustrating the values assumed by Cronbach's α at least for the mentioned groups ("Family/friends", "help of staff", "by themselves").
Minor comments
I have no minor comments to add.
Author Response
|
Reviewer 2 Comment 1: The manuscript is well written and illustrates an interesting research that could introduce some novelty in the field of palliative cares. |
Authors’ response 1: Thank you for these encouraging comments and your support in strengthening our manuscript. |
|
Reviewer 2 Comment 2: the literature examined is rather dated (only one third of the references has been published less than five years ago). |
Authors’ response 2: We have updated the manuscript to include newer references. A number of the references relate to the primary publications developing the outcome measures, consequently they will be older. |
|
Reviewer 2 Comment 3: Quantitative analyses could also be improved, as they are based only on descriptive statistics and results presented are not clear. As an example, it is not clear why absolute frequencies reported in table 1 concerning diagnoses, completion, and time of assessment do not sum to 1.507. |
Authors’ response 3: We have corrected and revised table 1. We have added descriptors of each percentage for AKPS
We have added a sentence in the text “For a more detailed account of the statistical analysis, interested readers are invited to refer to (Sandham et al, 2022).” Page 8 line 319 |
|
Reviewer 2 Comment 4: data illustrated in Table 2 are not commented on the main body of the manuscript, or, at least, it is not clear to me how they relate to the data illustrated in Table 3 and commented on lines 241-250. |
Authors’ response 4: We have added additional comments to direct the reader towards the table and expanded on aspects of the text. Tables 2 and 3, pages 7 and 8.
|
|
Reviewer 2 Comment 5: I could not find data concerning POI (lines 248-250) in Tables 1-2. |
Authors’ response 5: Table 1 and 2 do not report POI. We have modified the text to differentiate between the data that is reported in each table. The POI is reported in table 3 and comments in the text were added to highlight the POI. We also added the abbreviation POI onto the table to help make the data more visible.
|
|
Reviewer 2 Comment 6: Summary statistics illustrated in Table 3 are difficult to interpret, consequently Authors are invited to improve the interpretation of the data in the main body of the manuscript. As an example, why both St.Dev. e St.Error are reported? |
Authors’ response 6: We have added additional information interpreting the statistics and included some descriptors of what the scores mean from the IPOS and AKPS. We removed standard error for table 3.
|
|
Reviewer 2 Comment 7: the analysis of Cronbach’s α of the IPOS items (lines 271-273 could be integrated with a table illustrating the values assumed by Cronbach's α at least for the mentioned groups ("Family/friends", "help of staff", "by themselves"). |
Authors’ response 7: We thank the reviewer and have considered how we could incorporate the alpha’s into a table however this was not easily presented in columns and it remains in the text. We also felt that it was not worth a table in its own right as it is a short line of text. We have added a footnote to the table to say the alphas. |
Reviewer 3 Report
The only novel, positive result from this descriptive study was "The present study found that MBPC may in part, enable clinicians to break the ‘conspiracy of silence’ where neither patient nor clinician initiates discussion of sensitive matters." This may be true, and if so, would be important enough to include this intervention into general palliative medicine practice.
However, if there is data (not just descriptive/narrative) to support this assertion, I could not find it. It may BE there, and if it is, it needs to be highlighted more effectively.
I don't see where this article advances medical knowledge otherwise. The Māori data would have been interesting but this was inconclusive: "Although the present study sought information from clinicians on the appropriateness of the measures for Māori, we did not find enough evidence to support or discourage the use of these measures for Māori and Pacific people."
The title of the article is "Strengthening Community End-of-Life Care through Implementing Measurement-based Palliative Care." MBPC is absolutely a very important concept; without it, how can we know what we're even doing? The body of the article must match the title, or one must be changed. Emphasizing how you demonstrated the implementing MBPC actually strengthened community end of life care would improve this article greatly. Again, it may be there, but if the reader has to dig too deeply to find the meaning behind the article, they will likely abandon the effort.
thank you! MBPC is extremely important and you've done a great service. Now you just need to highlight the results if possible.
Author Response
|
Reviewer 3 Comment 1: the only novel, positive result from this descriptive study was "The present study found that MBPC may in part, enable clinicians to break the ‘conspiracy of silence’ where neither patient nor clinician initiates discussion of sensitive matters." This may be true, and if so, would be important enough to include this intervention into general palliative medicine practice. |
Authors’ response 1: thank you for this feedback, we have included a line in the conclusion to highlight this. “Using MBPC assists both clinicians and patients to discuss sensitive issues and break the conspiracy of silence where issues are not raised by either the patient or the clinician.” |
|
Reviewer 3 Comment 2: However, if there is data (not just descriptive/narrative) to support this assertion, I could not find it. It may BE there, and if it is, it needs to be highlighted more effectively. |
Authors’ response 2: Added a line that draws attention to this part of the results, within the discussion section “The theme of “making emotional wellbeing assessments routine” suggests that implementing MBPC may in part, enable clinicians to break the ‘conspiracy of silence’ where neither patient nor clinician initiates discussion of sensitive matters (Lemus-Riscanevo et al., 2019).” |
|
Reviewer 3 Comment 3: I don't see where this article advances medical knowledge otherwise. The Māori data would have been interesting but this was inconclusive: "Although the present study sought information from clinicians on the appropriateness of the measures for Māori, we did not find enough evidence to support or discourage the use of these measures for Māori and Pacific people." |
Authors’ response 3: This is a common problem in research which is not centered on indigenous populations. The study was led by the service itself and they wished to disseminate the findings to support other hospices that were required to implement measurement based care within New Zealand. We acknowledge that further research is needed to understand Maori and Pacific experiences, consequently future research will be directed towards understanding Maori and Pacific responses. We did not want to over-interpret the findings as some of the staff may not have been familiar with Maori and Pacific models. |
|
Reviewer 3 Comment 4: The title of the article is "Strengthening Community End-of-Life Care through Implementing Measurement-based Palliative Care." MBPC is absolutely a very important concept; without it, how can we know what we're even doing? The body of the article must match the title, or one must be changed. Emphasizing how you demonstrated the implementing MBPC actually strengthened community end of life care would improve this article greatly. Again, it may be there, but if the reader has to dig too deeply to find the meaning behind the article, they will likely abandon the effort. |
Authors’ response 4: Added :
“In summary, implementing MBPC strengthened community end-of-life care by: (a) instituting a set of internationally recognised measures and demonstrating their reliability, (b) reporting benchmark or reference scores for a large NZ community patient sample, and providing qualitative evidence that clinical staff found these measures acceptable, informative and able to improve aspects of care.” P 11, line 460 |
|
Reviewer 3 Comment 5: thank you! MBPC is extremely important and you've done a great service. Now you just need to highlight the results if possible. |
Authors’ response 5: Thank you for this encouragement and your support in strengthening our manuscript. |
Round 2
Reviewer 3 Report
thank you for such a quick turn-around!